# Relationship between Lower Urinary Tract Symptoms and Prostatic Urethral Stiffness Using Strain Elastography: Initial Experiences

**DOI:** 10.3390/jcm8111929

**Published:** 2019-11-09

**Authors:** Jong Kyou Kwon, Do Kyung Kim, Joo Yong Lee, Jong Won Kim, Kang Su Cho

**Affiliations:** 1Department of Urology, Severance Check-Up, Health Promotion Center, Severance Hospital, Seoul 04527, Korea; jkstorm@yuhs.ac; 2Department of Urology, Soonchunhyang University Hospital, Soonchunhyang University College of Medicine, Seoul 04401, Korea; dokyung80@hotmail.com; 3Department of Urology, Severance Hospital, Urological Science Institute, Yonsei University College of Medicine, Seoul 03722, Korea; joouro@yuhs.ac; 4Department of Urology, Gangnam Severance Hospital, Urological Science Institute, Yonsei University College of Medicine, Seoul 06273, Korea; doctor2play@yuhs.ac

**Keywords:** prostatic hyperplasia, lower urinary tract symptoms, elastography

## Abstract

We attempted to visualize the periurethral stiffness of prostatic urethras using strain elastography in the midsagittal plane of transrectal ultrasonography (TRUS) and to evaluate periurethral stiffness patterns in relation to lower urinary tract symptoms (LUTS). A total of 250 men were enrolled. The stiffness patterns of the entire prostate and individual zones were evaluated using strain elastography during a TRUS examination. After excluding 69 men with inappropriate elastography images, subjects were divided according to periurethral stiffness into either group A (low periurethral stiffness, *N* = 80) or group B (high periurethral stiffness, *N* = 101). There were significant differences in patient age (*p* = 0.022), transitional zone volume (*p* = 0.001), transitional zone index (*p* = 0.33), total international prostate symptom score (IPSS) (*p* < 0.001), IPSS-voiding subscore (*p* < 0.001), IPSS-storage subscore (*p* < 0.001), and quality of life (QoL) score (*p* = 0.002) between groups A and B. After adjusting for relevant variables, significant differences in total IPSS, IPSS-voiding subscore, and QoL score were maintained. Men with high periurethral stiffness were associated with worse urinary symptoms than those with low periurethral stiffness, suggesting that periurethral stiffness might play an important role in the development of LUTS.

## 1. Introduction

Benign prostatic hyperplasia (BPH) is known to induce bladder outlet obstruction (BOO) and cause lower urinary tract symptoms (LUTS) [1]. Prostate volume usually increases with age, and this increased volume is considered to be the most significant risk factor for LUTS and BPH progression [2,3,4]. However, many studies have indicated that associations between prostate volume and LUTS severity are relatively modest [5,6]. For instance, a considerable portion of patients experience bothersome LUTS without identifiably increased prostate volume [7,8]. Researchers have explored other potential risk factors of LUTS, including intravesical prostatic protrusion, presumed circle area ratio, prostate contour based on bladder neck variation, prostatic urethral angulation, and peripheral zone thickness [9,10,11,12,13].

Interestingly, several reports have shown that a fibrotic change in the prostate produces stiffer tissue and reduces elasticity and compliance, thus inducing LUTS [14,15,16,17]. Han et al. suggested that calcification and fibrotic changes result from inflammatory processes around the prostatic urethra and are independently associated with urinary flow rate and urinary symptoms [18]. In this respect, the evaluation of periurethral elasticity can be a promising approach to elucidate the pathophysiology of male LUTS. Meanwhile, elastography has been considered to be a non-invasive and cost-effective imaging technology for tissue elasticity [19,20]. Several studies have used elastography to examine the elastic properties of several organs, such as the breast, liver, and thyroid. However, only a few recent studies have used it to examine the elastic properties of the prostate [21,22,23]. In these studies, researchers focused on differences in elasticity in adenoma and the peripheral zone between BPH and non-BPH groups, reporting average stiffness measurements from random points in individual prostate zones. However, the overall elasticity of the prostate can vary greatly from zone to zone and from patient to patient. Furthermore, all previous studies were performed in transverse images only, which do not thoroughly reflect changes in elasticity stemming from prostatic fibrosis in the periurethral course, which has been suggested to be a novel cause of LUTS in patients with BPH [14,16,24,25]. In the present study, we attempted to visualize the periurethral stiffness of the prostatic urethra using strain elastography and to evaluate relationships between periurethral stiffness patterns and LUTS, using both transverse and midsagittal plane images.

## 2. Materials and Methods

### 2.1. Subject Selection and Medical Ethics

The medical records of men who had visited our clinic (Severance Check-Up, Health Promotion Center, Yonsei University Health System, Seoul, Korea) and underwent transrectal ultrasonography (TRUS) with strain elastography between August 2017 and May 2018 were retrospectively reviewed. The following eligibility criteria were used for subjects in the present study: (1) men aged 40–70 years, (2) a complete international prostate symptom score (IPSS) questionnaire, including quality of life (QoL) score, (3) no medical therapy for BPH, (4) no history of urogenital cancer, such as bladder or prostate cancer, (5) no uncontrolled diabetes mellitus, (6) no neurologic disease history that could influence LUTS, (7) no previous lower urinary tract surgery, (8) no evidence of urogenital infections, and (9) no urinary tract abnormalities. The Institutional Review Board of Gangnam Severance Hospital approved the study protocol (Approval No. 3-2018-0140).

### 2.2. Assessment of Prostatic Anatomical Factors Using TRUS and Elastography

TRUS with elastography was performed using a single BK 3000 scanner (BK Medical, Herlev, Denmark). All examinations were performed by an experienced urologist (J.K.K.) with five years of experience in performing TRUS and elastography examinations. For all subjects, a conventional TRUS examination was performed in the left decubitus position with the subjects’ buttocks located at the edge of the bed and their knees bent up to their chests. Total prostate volume (TPV) and transitional zone volume (TZV) measurements were investigated using the prostate ellipsoid formula (height × width × length × π/6). Transitional zone index (TZI) was calculated according to the formula TZI = TZV/TPV [2]. After finishing a conventional-scale TRUS examination, elastography was performed. The elastogram mapping images were merged onto gray-scale images to obtain elasticity scales with a color-coded image (blue for the hardest areas and red for the softest areas). Elastography was performed both on the largest transverse section and the midsagittal plane including the urethral course of the prostate. As in previous studies [26,27], movement of the transducer was repeated with different compression ratios until a constantly stable and reproducible elastogram image series was acquired. To prevent any obvious indentation when producing a flat prostate edge, the operator placed the least amount of pressure on the prostate while gently compressing the prostate edge. As shown in Figure 1, the elasticity of the periurethral area was displayed as colored scales according to the area’s relative stiffness degree (red for soft consistency, blue for hard consistency, and yellow/green for intermediate consistency), and we ensured that each elastogram was performed correctly with proper contact between the probe tip and the posterior aspect of the prostate by monitoring a direct feedback system (quality scale bar) for all cases. All obtained images during examinations were recorded on the internal hard disk of the ultrasound equipment.

### 2.3. Defining the Subject’s Groups According to Periurethral Elastography

In all subjects, elastography images merged with gray-scale images were obtained in transverse and midsagittal planes using TRUS. Then, all images were evaluated according to elastography patterns by two urologists who were blinded to the subjects’ clinical features, including age, International Prostate Symptom Score (IPSS), and prostate-specific antigen (PSA). When elastography images were inconsistent or of low quality due to several reasons in repeated examination, the subject’s case was excluded for statistical analysis. Subjects with properly obtained elastography images were divided into two groups: Group A was defined by an entirely delineated prostatic urethra with predominantly yellow to red signals, suggestive of high elasticity (low periurethral stiffness, soft periurethral consistency) in elastography. Therein, the stiffness of the entire prostatic urethra course could be distinguished from adjacent prostatic tissue. Group B was defined by proximal prostatic urethras that showed green to blue signals, suggestive of low elasticity (high periurethral stiffness, hard periurethral consistency), and their urethral stiffness was similar to the adjacent prostatic tissue (Figure 2). When the initial two investigators (J.K.K. and D.K.K.) had different opinions, another urologist (K.S.C.) reviewed the images and made a final decision for defining group allocation.

### 2.4. Statistical Analysis

All statistical analyses were performed on data of groups A and B, which showed a consistently similar pattern of periurethral elastography. Statistical comparisons of continuous variables between the two groups were conducted using Student’s *t*-test and the Mann–Whitney U-test. One-to-one propensity score matching was performed by matching the subjects in group A with those in group B. Propensity score matching was performed using subject age, TPV, TZV, and PSA. Total IPSS was classified into IPSS-post-micturition, storage (IPSS-S), and voiding (IPSS-V) symptom subscores [1,28]. These subscores were also analyzed separately. Two-sided tests were performed, and *p*-values less than 0.05 were considered to be statistically significant.

All computations were conducted using IBM SPSS Statistics (Version 25.0. Armonk, NY: IBM Corp, USA) and R freeware v3.5.3 (R Foundation for Statistical Computing, Vienna, Austria; http://www.r-project.org); the MatchIt package v3.0.2 (https://www.rdocumentation.org) was used to perform propensity matching.

## 3. Results

The medical records of 250 consecutive men were reviewed, and 69 subjects were excluded due to inappropriate periurethral elastography (shown in Table 1 and Figure 3). The data of 181 subjects were used in the final analysis. According to periurethral stiffness patterns, 80 men were classified into group A, and 101 men into group B. The median age (interquartile range) for these subjects was 53.0 years (49.0–58.0), and their median PSA (interquartile range) was 0.98 ng/mL (0.60–1.20). TPV, TZV, and TZI values were 26.50 ± 6.86 mL, 8.25 ± 5.33 mL, and 0.31 ± 0.28, respectively. Total IPSS, IPSS-V, IPSS-S, IPSS-post-micturition, and QoL scores were 9.46 ± 5.01, 4.52 ± 2.74, 3.63 ± 2.24, 1.31 ± 1.13, and 2.09 ± 1.08, respectively.

In most subjects, the overall elastography pattern was heterogeneous in the entire prostate and within each anatomical zone (Figure 1). In addition, the peripheral zone showed relatively higher stiffness than the transitional zone. According to the periurethral stiffness pattern based on the elastogram findings, subjects were divided into two groups. The images of group A (low stiffness, soft consistency) consistently showed a yellow to red signal along the urethral course from the bladder neck to the verumontanum and a periurethral area easily distinguishable from the prostatic tissue. The images of group B (high stiffness, hard consistency) consistently showed a green to blue signal along the urethral course, with periurethral stiffness similar to the transitional zone (Figure 2).

In the comparison analysis, group A showed significantly lower total IPSS, IPSS-V, IPSS-S, and QoL than group B. In addition, group B was significantly older, with greater TZV and TZI than group A. There were no significant differences in TPV, PSA, and IPSS-post-micturition subscores between the two groups (Table 2). Group B showed more symptomatic subjects than group A, with a higher proportion in group B exhibiting moderate and severe IPSS (Figure 4).

Propensity score-matching analysis was performed to adjust for subjects’ age and prostate volumes, and 140 subjects (70 patients for each group) were selected (Table 2). In the matched group comparison, no differences in IPSS-S and IPSS-post-micturition were noted. However, significant differences in total IPSS, IPSS-V, and QoL were recorded. The proportions of symptomatic subjects were maintained (Figure 4).

## 4. Discussion

Previous studies have reported that patients with BPH exhibit significant differences in smooth muscle cells, including elastic and collagen fibers with a higher rate of stromal hyperplasia than of epithelial hyperplasia, especially in symptomatic patients [29,30]. Pathologic hypertrophy of BPH nodules presumably occurs first in the periurethral zone, followed by enlargement of the entire glandular nodule, resulting in BOO and LUTS. Therefore, prostatic fibrosis, especially in the periurethral area, has been suggested to be a novel cause of LUTS in patients with BPH [14,16,24,25]. Researchers have focused on the periurethral area as one of the important regions for determining symptom severity. Cantiello et al. [14] analyzed periurethral tissues obtained from radical prostatectomies and suggested that fibrotic changes in the periurethral area induced by the inflammatory process promoted urethral stiffness, resulting in LUTS. After examining periurethral tissues obtained from radical prostatectomy, Ma et al. [17] also reported that symptomatic patients showed significantly higher stiffness of periurethral tissue than asymptomatic patients. They also found higher collagen content and lower glandularity in symptomatic patients. Based on these studies, periurethral fibrosis and stiffness have been considered causes of decreased urethral flexibility that compromise the ability of the prostatic urethra to dilate to adequately accommodate urinary flow during micturition.

A few recent studies have reported a relationship between LUTS/BPH and prostate elasticity that was evaluated using TRUS with elastography, which is a novel medical imaging modality that maps the elastic properties and stiffness of soft tissue. Alan et al. [31] revealed that patients with BPH showed higher stiffness in the central zone than healthy volunteers, but found no significant difference in the peripheral zone. The elasticity of the peripheral zone was lower than that of the central zone. Similarly, Zhang et al. [23] reported a higher stiffness for the inner gland zone than the outer gland zone of the prostate in patients with BPH, as well as a significantly higher average stiffness in patients with BPH than those with a normal prostate. The most striking findings of these studies were the significant associations between the elasticity difference of the prostate zonal area and BOO in patients with LUTS/BPH. The elastic modulus of the transitional zone was the indicator most strongly correlated with BOO stage, suggesting that the elasticity of the transitional zone can be used as a reliable non-invasive indicator of BOO.

In the present study, unlike previous studies that used transverse images only, sagittal images were also evaluated for visualization of periurethral stiffness patterns along the entire urethral course. To the best of our knowledge, this is the first study to evaluate the relationships between LUTS/BPH and the entire course of periurethral elasticity in sagittal images from prostatic ultrasound using elastography. Previous researches have mainly focused on the average stiffness of each zonal anatomy and performed elastography in the transverse image at a random location of the prostatic urethra [23,31]. As shown in Figure 1, the overall elasticity pattern of each zona showed heterogeneous and gradual stiffness patterns, in which the representative value for average elasticity for each zone from random point measurement can be varied in the repeated measurements. Furthermore, previous approaches were not enough to reflect the gradual changes of elasticity from prostatic fibrosis in the periurethral area. In addition, in contrast to previous studies that used shear-wave elastography, our study used real-time quasi-static elastography [23,26,31]. Although this type of elastography obtains real-time images from a wide area much faster, it only provides relative and qualitative results as comprehensive image patterns within the region of interest; thus, we could not perform quantitative comparisons as reported in previous studies. However, as shown in Figure 1, the intraprostatic zonal elasticity was quite heterogeneous in the central and peripheral zones, especially in the periurethral area. In this regard, approaches of previous studies that used the average elasticity value from a random point may not properly represent the overall zonal elasticity. In addition, the shear-wave elastography ultrasound, which previous studies used, is not able to provide continuous visualization of periurethral elasticity along the entire urethral course at a glance; it also requires time-consuming processing and captures only a limited region of interest. Considering the complexity of the intrazonal anatomy, quasi-static elastography may have some advantages in clinical practice to evaluate the overall elasticity of the prostate.

In this study, we investigated the elasticity pattern in the periurethral area and found that this pattern is associated with LUTS. This finding suggests that periurethral stiffness is an important etiology for understanding the pathophysiology of male LUTS. We also performed a propensity score matching analysis to more clearly elucidate the relationship between urethral stiffness and urinary symptoms. However, our study did not provide enough information to clarify the mechanism underlying the relationship between elasticity and BOO. Firstly, our study was based on health check-up data in the health promotion center, and asymptomatic or mild symptomatic individuals were also included. Additional works in real clinical settings are needed to elucidate the role of periurethral stiffness in LUTS/BPH. Furthermore, despite several positive reports supporting the relationship between the changes in periurethral stiffness and the developing of LUTS/BPH, it is not yet clear that there is a definite mechanism underlying the relationship between urethral stiffness and urinary symptoms.

Secondly, our study used a different type of elastography than that used in previous studies [23,31]. Our comprehensive but qualitative periurethral stiffness patterns are worth being reevaluated using quantitative shear-wave elastogram techniques. We suggest that if periurethral stiffness could be a factor in the development and progression of LUTS/BPH, quantitative changes might have some usefulness for understanding the pathophysiology of LUTS/BPH. Thirdly, for our study subjects, no data were available on objective urodynamic parameters such as peak flow rate, post-void residue, and BOO index. We suggested that our initial experiences with visualization of periurethral stiffness with strain elastography need to be validated in regards to whether observed differences in periurethral stiffness patterns hold significant relationships with other objective parameters of the pathophysiology of LUTS/BPH.

## 5. Conclusions

The stiffness pattern of the prostate is heterogeneous throughout the entire prostate and even within each zone. The peripheral zone tends to show relatively higher stiffness than the transitional zone in patients with LUTS/BPH. Strain elastography is a feasible and useful method with which to visualize periurethral stiffness patterns of the entire prostatic urethra during TRUS. Notably, men with high periurethral stiffness in the proximal urethral course appear to show worse urinary symptoms than those with low periurethral stiffness. This finding suggests that gradual change in periurethral elasticity can be an important indicator in the development of LUTS.

## Figures and Tables

**Figure 1 jcm-08-01929-f001:**
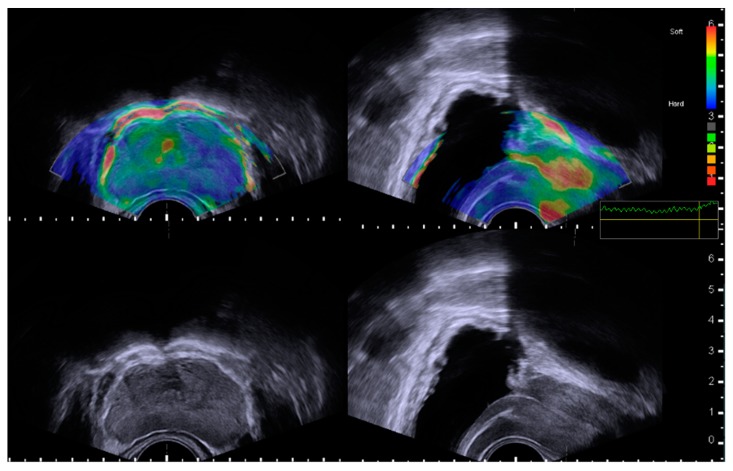
Measurement of prostate elastography. Elastogram-merged mapping images to the gray-scale images delineate elasticity scales with a color-coded image according to the continuous elasticity color scale (blue for the hardest area and red for the softest area). Elastography was performed on the largest transverse section (upper left) and the midsagittal plane including the entire urethral course of the prostate (upper right). Quality-scale monitoring was performed for each examination to ensure whether the elastography was performed correctly, with proper pressure and contact between the probe tip and the posterior aspect of the prostate, or not.

**Figure 2 jcm-08-01929-f002:**
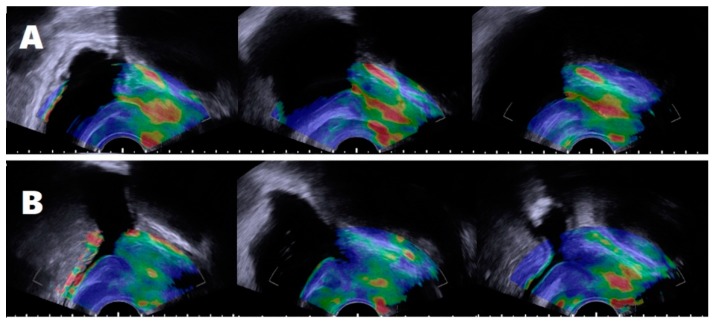
Two patterns of periurethral stiffness in the sagittal planes. After repeated elastography, the images of group A showed a yellow to red signal along the urethral course from the bladder neck to verumontanum (**A**) and a periurethral area easily distinguishable from the prostatic tissue. Images of group B showed a green to blue signal in the proximal prostatic urethral course (**B**), with periurethral stiffness similar to the transitional zone.

**Figure 3 jcm-08-01929-f003:**
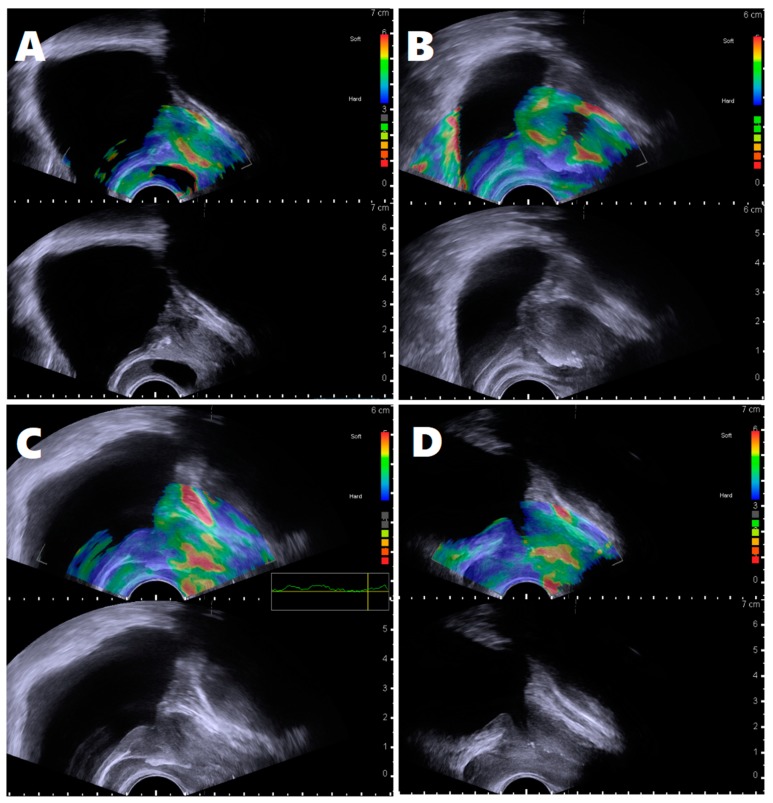
The reasons for inappropriate periurethral elastography. (**A**) midline cyst or periurethral degenerative cystic nodule; (**B**) prostate calcification including periurethral calcification; (**C**) inappropriate quality-scale bar which failed to reach a satisfactory quality scale; (**D**) lacking echotexture of bladder neck and proximal urethral course in B-mode.

**Figure 4 jcm-08-01929-f004:**
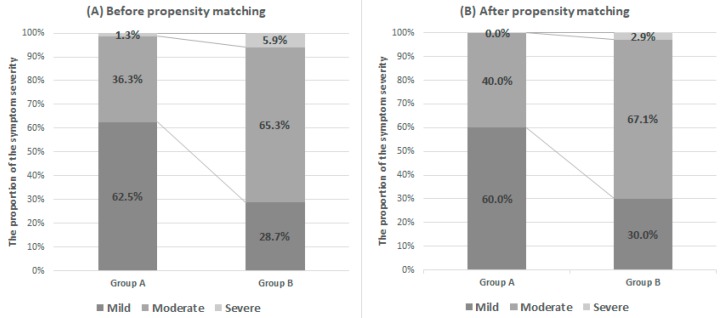
Distribution of subjects according to symptom severity using IPSS in each group before (**A**) and after (**B**) propensity score matching. In the IPSS system, the banding of symptom severity was set as 1–7 mild, 8–19 moderate, and 20–35 severe to categorize patient symptoms and to help physicians manage treatment.

**Table 1 jcm-08-01929-t001:** Cases of inappropriate periurethral elastography and reasons.

Midline Cyst or Periurethral Degenerative Cystic Nodule	4
Prostate calcification including periurethral calcification	24
Inappropriate quality-scale bar	14
Lacking echotexture of bladder neck and proximal urethral course in B-mode	27

**Table 2 jcm-08-01929-t002:** Comparison of prostatic parameters and urinary symptoms by periurethral stiffness before and after propensity score matching.

**Before Matching**	**Group A (*N* = 80)**	**Group B (*N* = 101)**	***p*-Value**
Age (y)	51.5 (47.0–56.0)	54.0 (50.0–59.0)	0.022
PSA (ng/mL)	0.949 (0.60–1.10)	1.004 (0.60–1.20)	0.735
TPV (mL)	25.54 ± 6.49	27.27 ± 7.08	0.092
TZV (mL)	6.76 ± 2.91	9.44 ± 6.42	0.001
TZI	0.26 ± 0.06	0.35 ± 0.37	0.033
IPSS			
Total	7.38 ±4.42	11.12 ± 4.85	<0.001
IPSS-V	3.18 ± 2.63	5.62 ± 2.98	<0.001
IPSS-S	3.01 ± 2.17	4.12 ± 2.18	<0.001
Post-micturition	1.19 ± 0.91	1.38 ± 1.87	0.074
QoL	1.81 ± 1.10	2.31 ± 1.01	0.002
**After Matching**	**Group A (*N* = 70)**	**Group B (*N* = 70)**	***p*-Value**
Age (y)	52.0 (48.5–58.0)	52.5 (48.0–57.0)	0.660
PSA (ng/mL)	0.89 (0.60–1.10)	0.96 (0.60–1.20)	0.869
TPV (mL)	25.46 ± 5.76	24.97 ± 54.04	0.596
TZV (mL)	6.99 ± 2.90	7.01 ± 2.74	0.952
TZI	0.27 ± 0.62	0.28 ± 0.60	0.509
IPSS			
Total	7.66 ± 4.36	10.51 ± 4.68	<0.001
IPSS-V	3.24 ± 2.20	5.31 ± 2.73	<0.001
IPSS-S	3.14 ± 2.17	3.61 ± 2.04	0.187
Post-micturition	1.27 ± 0.95	1.59 ± 1.21	0.089
QoL	1.84 ± 1.14	2.24 ± 1.04	0.032

International Prostate Symptom Score (IPSS), IPSS storage subscore (IPSS-S), IPSS voiding subscore (IPSS-V), total prostate volume (TPV), transitional zone volume (TZV), transitional zone index (TZI), prostate-specific antigen (PSA), quality of life (QoL).

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
