# Peer review of "Relationship between Lower Urinary Tract Symptoms and Prostatic Urethral Stiffness Using Strain Elastography: Initial Experiences"

_jcm, 2019, doi:10.3390/jcm8111929_

Round 1

Reviewer 1 Report

In this study elastography is performed during transrectal ultrasound of the prostate.

The authors found a relation between stiffness of the prostate o and age and IPSS in middle aged men.

In my opinion it is a well designed study. Although quite a large group of patients had to be excluded, a significant number of men remain.

In figure 2, showing a patient of group B (stiff prostate), the colours indicating stiffness are not so clear, i.e. the distal prostatic urethra is red.

My main concern is in the conclusion. I do agree with the statement that elasticity of the prostate is related to symptoms, which can be promising to elucidate the pathophysiology of LUTS. However, it is not useful for monitoring response to treatment, because that should be done with IPSS or PROMS or whatever. It is also not useful to use elasticity to predict symptoms. Why should you do TRUS and elastography on a patients with no symptoms (yet)?

Author Response

In this study elastography is performed during transrectal ultrasound of the prostate. The authors found a relation between stiffness of the prostate o and age and IPSS in middle aged men. In my opinion it is a well-designed study. Although quite a large group of patients had to be excluded, a significant number of men remain.

Comment 1: In figure 2, showing a patient of group B (stiff prostate), the colours indicating stiffness are not so clear, i.e. the distal prostatic urethra is red.

Answer for comment 1: Thank you for your great comment. We completely agree with your opinion. As we mentioned in the manuscript and legend of figure 2 as ‘the urethral course from the bladder neck to verumontanum’, we tried to focus on the stiffness change in the ‘proximal prostatic urethral course’ which lies in inner gland zone. Previous studies also reported that patients with BPH showed higher stiffness in the central zone than healthy volunteers but found no significant difference in the peripheral zone [1,2]. These findings are compatible with our results. However, as you pointed out, our description was not so clear enough to convey our findings in the manuscript and figure legend. Thus, we revised the manuscript as follows:

# On page 3 of 11, line 112 to 115

Group B was defined by proximal prostatic urethras that showed green to blue signals suggestive of low elasticity (high periurethral stiffness, hard periurethral consistency), and their urethral stiffness was similar to adjacent prostatic tissue (Figure 2).

# On page 8 of 11, line 260 to 262 (Conclusion)

Notably, men with high periurethral stiffness in proximal urethral course appear to show worse urinary symptoms than those with low periurethral stiffness.

# Figure legend 2

Images of group B showed a green to blue signal in the proximal prostatic urethral course (B), with periurethral stiffness similar to the transitional zone.

Comment 2: My main concern is in the conclusion. I do agree with the statement that elasticity of the prostate is related to symptoms, which can be promising to elucidate the pathophysiology of LUTS. However, it is not useful for monitoring response to treatment, because that should be done with IPSS or PROMS or whatever. It is also not useful to use elasticity to predict symptoms.

Answer for comment 2: Thank you for your comment. We agree with your opinion. We suggested that the periurethral stiffness might be one of an indicator can predict treatment result based on the report from De Assiss et al [3]. They reported that the patients group who underwent selective artery embolization for treatment of BPH showed lower stiffness in the central zone by 19.0-29.8%and lower TZI by 45.0% in 1 month, and significant improvement of IPSS, QOL, prostate volume, Qmax, and PSA in all patients. They suggested that the embolization can block the influx of such mediators to the prostate, leading to the overall hypo-activation of the alpha-receptors. In addition, embolization leads to ischemic necrosis of transitional zone tissue, which could reduce alpha-adrenergic receptors density, also contributing to the reduction of prostate tone and the stiffness of the central zone. However, as you pointed out in the comments, the direct relationship between the change of stiffness in adenoma from the treatment or intervention and the change of proximal periurethral stiffness in our study is still not clear. Thus, we removed the descriptions in the discussion as follows:

# On page 8 of 11, line 247 to 249

We suggested that if the periurethral stiffness could be a part of the factors for development and progression of LUTS/BPH, quantitative changes might have some usefulness for understanding the pathophysiology of LUTS/BPH. or for monitoring response after medical treatment.

Comment 3: Why should you do TRUS and elastography on a patient with no symptoms (yet)?

Answer for comment 3: Thank you for your great comments. Our study was performed as a part of health check-up in the health promotion center, and asymptomatic or mild symptomatic individuals were also included. Accordingly, the mean total IPSS was relatively low considering the median age of 50s as shown in table 1. We revised the discussion section as follows:

 # On page 8 of 11, line 238 to 244

Firstly, our study was based on health check-up data in the health promotion center, and asymptomatic or mild symptomatic individuals were also included. Additional works in the real clinical settings is needed to elucidate the role of periurethral stiffness in LUTS/BPH. Furthermore, despite several positive reports supporting the relationship between the changes in periurethral stiffness and the developing of LUTS/BPH, it is not yet clear the definite mechanism underlying the relationship between urethral stiffness and urinary symptoms.

Reviewer 2 Report

The authors explain their investigation about the elasticity pattern in the periurethral area and its realtionship with LUTS.

The main commentary is about the sciencitifcal soundness about this experimental investigation. In one hand, it is not clear the mechanism underliying relationship urethral stiffness and urinary symptoms, and it´s important to clarify it in order to develop fuhter strategies in this area. In the other hand, the clinical application of this relationship is weak and probably doesn´t impact in clinical management.

Author Response

The authors explain their investigation about the elasticity pattern in the periurethral area and its realtionship with LUTS.

Comment 1: The main commentary is about the sciencitifcal soundness about this experimental investigation. In one hand, it is not clear the mechanism underliying relationship urethral stiffness and urinary symptoms, and it´s important to clarify it in order to develop fuhter strategies in this area. In the other hand, the clinical application of this relationship is weak and probably doesn´t impact in clinical management.

Answer for comment 1: Thank you for your great comment. We completely agree with your opinion that it is not clear the mechanism underlying relationship urethral stiffness and urinary symptoms yet, and the clinical application of this relationship is weak. As we described in the manuscript, there has been only a small number of studies on this matter. However, all of the previous studies have consistently suggested that the change of stiffness in adenoma and periurethral area from either physiologic changes or inflammatory process (e.g. calcifications) might have an important role in development of LUTS/BPH [4][5-9]. It supports that the changes of periurethral stiffness or central zonal elasticity might be a possible candidate affecting LUTS/BPH as we described in this work. We revised the discussion section as follows:

# On page 8 of 11, line 238 to 244

Firstly, our study was based on health check-up data in the health promotion center, and asymptomatic or mild symptomatic individuals were also included. Additional works in the real clinical settings is needed to elucidate the role of periurethral stiffness in LUTS/BPH. Furthermore, despite several positive reports supporting the relationship between the changes in periurethral stiffness and the developing of LUTS/BPH, it is not yet clear the definite mechanism underlying the relationship between urethral stiffness and urinary symptoms.

Round 2

Reviewer 2 Report

No additional comments